# Mast Cells Are Activated in the Giant Earlobe Keloids: A Case Series

**DOI:** 10.3390/ijms231810410

**Published:** 2022-09-08

**Authors:** Yukari Nakajima, Noriko Aramaki, Nao Takeuchi, Ayumi Yamanishi, Yoshiko Kumagai, Keisuke Okabe, Tomoaki Yokoyama, Kazuo Kishi

**Affiliations:** 1Department of Plastic and Reconstructive Surgery, Keio University Hospital, Tokyo 160-8582, Japan; 2Department of Dermatology, Shimizu Hospital, Shizuoka 424-8636, Japan

**Keywords:** keloids, mast cell degranulation, obesity, inflammation, fibroblasts, fibrosis, collagen

## Abstract

Mast cells and inflammatory cells are abundant in keloid and hypertrophic scar tissues. Even if the cause of physical injury is similar, such as piercing or scratching with hands, clinical findings show differences in the size of keloids in the same area. Hence, we performed histological studies on giant keloids larger than the earlobe, and other smaller keloids. We also examined the risk factors associated with the formation of giant lesions. No statistically significant differences in the association of the risk factors were observed. However, histological observations clearly showed a high number of degranulated or active mast cells with a trend towards a greater number of degranulated mast cells in the giant keloid tissues. Collagen production also tended to increase. Two patients with giant keloids were severely obese, suggesting that the persistent inflammatory state of obesity may also be involved in the growth of keloid lesions.

## 1. Introduction

The pathogenesis of keloids, namely, excessive growth with more pain and itching than normal scar tissue reformation, remains unclear. Conservative treatment methods for keloids include compression therapy, topical steroidal injections, and topical steroids application. The most invasive treatment option is a combination of surgery and radiotherapy. However, keloids may recur even after excision of the lesion and post radiotherapy [1], and no definitive treatment plan has yet been developed.

Chronic inflammatory changes persist in the reticular dermis of the keloid tissues [2]. There are several reports which focused on some cell types within the keloid tissue, but, to date, there are no reports that compared cell types with size of the keloid [3,4]. Hence, in this study, we aimed to compare the mast cell activation with the size of the keloid lesion.

## 2. Results

We studied five cases of keloids in the earlobe. Patient C and D presented with ‘giant keloids’ that exceeded the size of their earlobe (Figure 1). Injuries due to excessive scratching was the cause of keloid formation in patient D, whereas injuries from piercing was the cause in the other patients.

The patients were diagnosed with ‘keloids’ by hematoxylin and eosin staining (Figure 2I). More specific histological examination with toluidine blue staining revealed activation of the mast cells in the keloids (Figure 2II).

Collagen levels are known to be positively correlated with activated mast cells (Table 1 and Table 2).

The excised tissues were positive for degranulated mast cells, which were identified using toluidine blue staining, as shown in Figure 3. Additional immunohistochemistry staining with Tryptase and CD117 revealed a similar number of mast cells in the 1 mm^2^ areas scanned (Figure 2III,IV, Table 2). Though no statistically significant differences were observed; degranulated mast cells were more common in giant keloid lesions (Figure 4, Table 2).

No statistically significant differences were observed for these factors (Table 3 and Table 4). We also found that collagen production by fibroblasts was increased in tissues with more degranulated mast cells, as indicated by the size of the keloid tissue in the physical examination (Figure 1 and Figure 2I).

## 3. Discussion

In general, there are more activated degranulated mast cells in keloids than in normal skin. Mast cells usually produce a degranulation reaction that transports secretory granules to the cell surface upon binding to specific IgE antibodies, and release chemical mediators, such as histamine, in the granules. These mediators are known to promote a local inflammatory response, which also promotes fibroblast activation and the synthesis of the extracellular matrix, leading to fibrosis [5].

Specifically, heparin, when released from mast cells, assists in the binding of fibronectin to collagen and activates fibroblasts by retaining basic fibroblast growth factor. Tryptase in the granules also increases collagen type 1 production in fibroblasts. Transforming growth factor β, a cytokine central to fibrosis, is thought to be responsible for the transformation of fibroblasts into myofibroblasts. Furthermore, it has been reported that co-culture with mast cells markedly promotes the contractility of fibroblasts [6]. Histamine and interleukin-4 (IL-4) also increase fibroblast migration. It is also characterized by the storage of tumor necrosis factor α, especially in granules, which can be released early in the process [7].

In addition, it has been reported that the release of histamine from mast cells is accelerated by the administration of protamine to rats and that the number of collagen fibers produced was significantly higher than that in the non-treated group [8]. In other words, the more degranulated the mast cells, the higher the proliferation of keloid lesions. This was also observed in our study where we found the giant keloid lesions had many degranulated and activated mast cells, which might have contributed to the number of collagen fibers produced.

Some stem cell factors secreted by fibroblasts stimulate mast cells, and their interrelationships are very close [9]. Studies have reported that mast cell activation also promotes the differentiation of neutrophils into Th2 cells by releasing histamine, IL-4, IL-13, proteases, and other substances. Thus, chronic inflammatory activation is manifested as repeated acute inflammation [10].

Furthermore, while some researchers have reported that there is no association between obesity and mast cells, it has recently been reported that the adipose tissue of obese humans and mice contains more mast cells than that of the lean group. The stabilization of mast cells in obese mice has been reported to reduce the expression of inflammatory cytokines and other factors and improve glucose homeostasis and energy expenditure [11]. In other words, mast cell activation is likely to be elicited in obese patients leading to an increased number of mast cells, as obesity is also a persistent inflammatory condition.

Although no statistically significant differences were observed in this study, it was observed that giant keloid lesions were formed in two of the cases who can be considered obese based on the World Health Organization (WHO)-defined maximum body mass index (BMI). Hence, we suggest that there might be an association between obesity and mast cell activation in keloid lesions.

Mast cells play a prominent role in defense mechanisms and are essential for biological defense; however, in keloids, therapies that locally suppress this degranulation response might be effective. It is possible that, under chronic inflammation, a specific fibroblast phenotype continues to stimulate mast cells. We plan to pursue this therapeutic development, including additional single-cell analysis experiments to investigate the factors that contribute to mast cell activation in wound healing.

## 4. Materials and Methods

We performed excision of earlobe keloids for five patients at our institution between June 2020 and February 2022. Excision was performed under local anesthesia and, except in one case, the postoperative wound was treated with electron beam therapy. Pathological examination of the excised lesions was performed, and the pathologists confirmed the diagnosis as ‘keloid’ in all the cases.

### 4.1. Histopathology

The keloid tissues were fixed in 10% paraformaldehyde and embedded in paraffin. For histology, the embedded tissue sample was sliced into 7 µm sections. The sections were stained by hematoxylin & eosin, and the histological examinations were performed by dermatologists and plastic surgeons. Toluidine blue dye was used to study the mast cell degranulation in the giant keloids which exceeded the size of the earlobe and tended to accumulate mast cells.

### 4.2. Immunohistochemistry

To identify mast cells more precisely, we also performed tryptase and CD117 staining.

After deparaffinization and rehydration, the tissue sections were treated with antigen retrieval buffer (Leica Biosystems, Newcastle Upon Tyne, UK) at 100 °C for 20 min, followed by washing with double distilled water. Tissue sections were incubated with antibody against mast cell tryptase (Leica Biosystems, Newcastle Upon Tyne, UK)) or anti-human CD117 (c-kit, Dako, Carpinteria, CA, USA) for 15 min, post primary antibodies (Leica Biosystems, Newcastle Upon Tyne, UK)) for 8 min, followed by secondary antibody HRP-conjugated anti-rabbit IgG (Leica Biosystems, Newcastle Upon Tyne, UK)) for 8 min. After washing, sections were treated with 4% hydrogen peroxide, and 3,3′-daminobenzidine tetra-hydrocholoride (Leica Biosystems, Newcastle Upon Tyne, UK)) for 10 min, counter-stained by Mayer’s hematoxylin (Leica Biosystems, Newcastle Upon Tyne, UK)), dehydrated, and mounted with Entellan™ new (Merck KGaA, Darmstadt, Germany). All images were obtained with Nano zoomer (Hamamatsu Photonics, Hamamatsu, Japan).

### 4.3. Visual Collagen Quantitations and Statistical Analysis

We quantitated the collagen levels in keloids by Image J software. We equalized the color contrast, changed to gray scale, and analyzed the collagen-containing areas

We randomly selected three areas of 1 mm^2^ in the dermal mesenchyme in each case and counted the cells visually.

We have collected the demographic and risk factor data of the patients (Table 3). Fisher’s exact test was performed to study the association between the categorical variables. *p* value was considered to be significant at *p* ≤ 0.05.

## 5. Conclusions

Although there were no statistically significant differences in the factors associated with giant keloid lesions in this study, there were clear differences in the histological observations regarding mast cell degranulation.

## Figures and Tables

**Figure 1 ijms-23-10410-f001:**
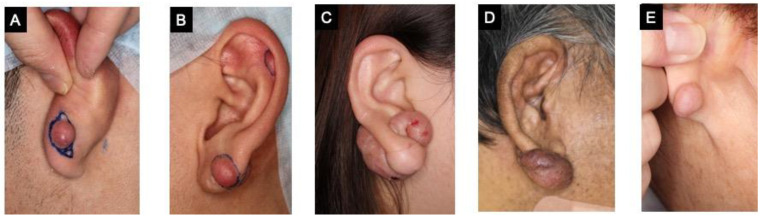
Patients with keloids in the earlobe. (**A**–**E**) were shown in order of increasing age.

**Figure 2 ijms-23-10410-f002:**
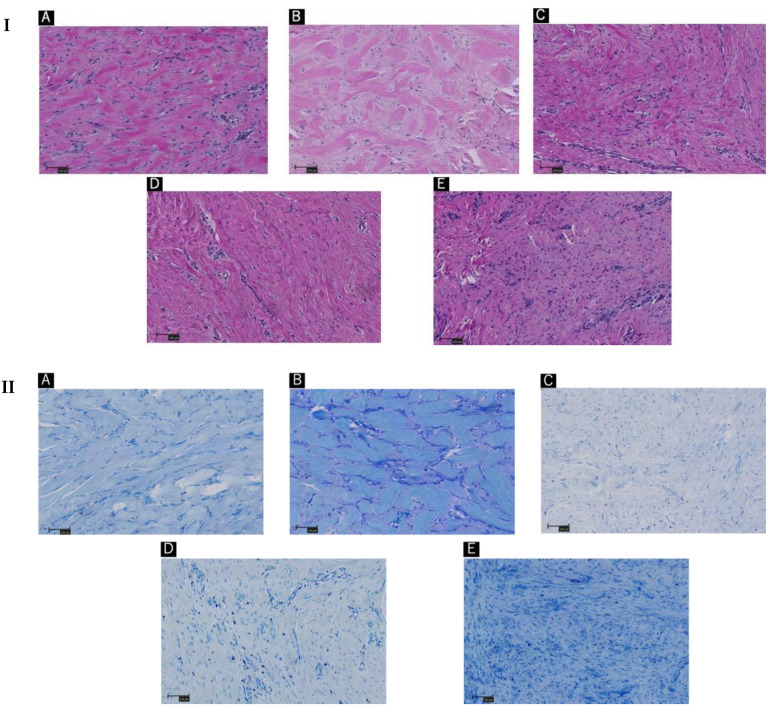
(**A**–**E**) indicates Patients’ order in Figure 1. (**I**) Hematoxylin and eosin staining of keloid tissue in the dermal mesenchyme specimens of the patients. The order in the same sequence as Figure 1. (**II**) Toluidine blue staining of keloid tissue in the dermal mesenchyme specimens of the patients. (**III**) Immunohistochemistry with tryptase staining. (**IV**) Immunohistochemistry with CD117 staining. Scale bar = 100 µm.

**Figure 3 ijms-23-10410-f003:**
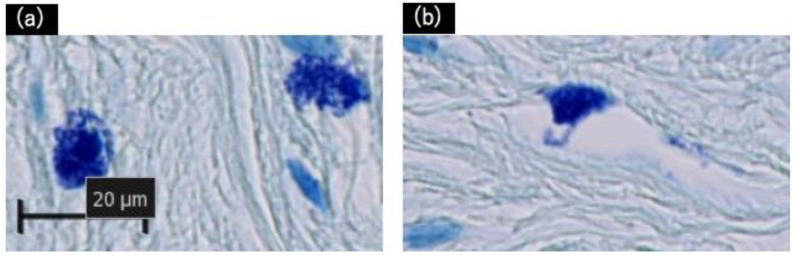
Histological validation of degranulation from mast cells with toluidine blue staining. (**a**) Positive: Mast cells have distinctly degranulated particles. (**b**) Negative: Granulation is not clearly observed. Scale bar = 20 µm.

**Figure 4 ijms-23-10410-f004:**
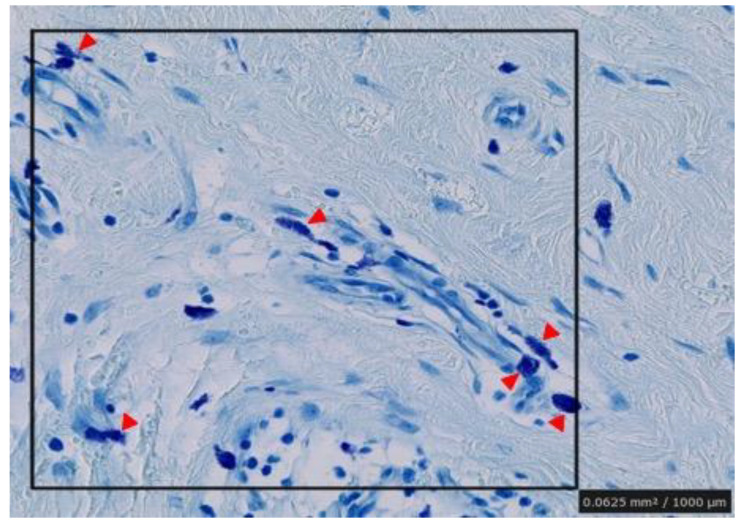
Counting of number of degranulated mast cells in random areas of the specimens. 1 mm^2^ areas were selected in the dermal mesenchyme and the cells were counted visually in each section. Arrowheads indicate positive mast cells.

**Table 1 ijms-23-10410-t001:** The amount of collagen in the lesion (%).

Patient A	Patient B	Patient C	Patient D	Patient E
50.68	48.20	69.12	58.85	58.34

Collagen level was quantitated visually in the scars by Image J software (version 2.1.0/1.53c, U.S. National Institutes of Health, Bethesda, MD, USA). These were equalized by color contrast, changed to gray scale, and collagen was analyzed in the area.

**Table 2 ijms-23-10410-t002:** The number of mast cells in each histochemical section.

	Patient A	Patient B	Patient C	Patient D	Patient E
Tryptase + cells	5.3	5	9	9	8.7
CD117 + cells	4.3	4	7	7.3	7.7
Activated mast cells	3	2	5	5.3	3.7

Numbers indicate the average of counted cells in three areas for each following Figure 3 and Figure 4. ‘Giant lesion’ tend to have more mast cells and its positive ratio is higher than the others.

**Table 3 ijms-23-10410-t003:** Demographic and risk factor data of the study participants.

No.	Age (Years)	Sex	BMI *	Lesion Side	History of Piercing	History of Pruritus	Duration of Keloid Appearance (Months)	Past Medical History
Patient A	21	Male	20.45	Left	+	-	10	N/A
Patient B	22	Male	19.57	Left	+	-	18	N/A
Patient C	32	Female	54.11	Right	+	+	108	N/A
Patient D	43	Male	48.98	Right	-	+	60	Schizophrenia
Patient E	73	Female	22.64	Left	+	-	24	Bronchiectasis

* Body mass index (BMI) was calculated based on World Health Organization (WHO) standards, ‘+‘ indicates presence, ‘-‘ indicates absence.

**Table 4 ijms-23-10410-t004:** Fisher’s Exact test for the possible risk factors.

Risk Factor	N *	*p* Value
Lesion side	2	0.1
BMI (>25)	2	0.1
History of piercing	1	0.4
History of pruritus	2	0.1
Duration of keloid appearance (>60 months)	2	0.1
Number of degranulation mast cells (>5)	2	0.1

* N implies giant lesion.

## Data Availability

Not applicable.

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
