# Peer review of "Mast Cells Are Activated in the Giant Earlobe Keloids: A Case Series"

_ijms, 2022, doi:10.3390/ijms231810410_

Round 1

Reviewer 1 Report

Manuscript ID: ijms-1821861

Title: Histological study of giant keloids and mast cells in the earlobe

1.      The title needs to be improved, and the present one tells the readers a general study of giant keloids and mast cells in the earlobe, and this is too general, and the improved one should contain the major finding of this study.

2.      Table1. The current number of activated mast cells in subject 4 and 5 are 5.3 and 3.7. Are these values the average number of activated mast cells in a certain field under the microscope? This should be mentioned clearly in the table note.

3.      Table 3, what is the meaning of n in the table?  This is very hard to understand.

4.      What is the merit and limitation of this study?

5.      The authors indicated that there might be association between obesity and mast cell activation. How to explain such association?

6.      The reviewer suggests that this manuscript is written in the case report format.

7.      The number of “Institutional Review Board Statement” must be provided in the manuscript.

Reviewer 2 Report

This is an interesting study on the histological analysis mast cells in the giant keloids. However, several issues need to be addressed:

1) Description of the control groups is missing. The non-keloid controls also need to be included to strengthen this study.

2) Immunohistochemistry with antibodies labeling the specific markers of mast cells should be performed.

3) Scar assessment, especially the histologically quantitative measurements, should be conducted and correlated with the mast cell analysis data in these tissues.

4) Figure legends seem missing.

5) The section of Materials and Methods needs detailed information.

Round 2

Reviewer 1 Report

My previous questions and comments were replied, and the IRB number remains not provided. This should be provided before publication. 

Reviewer 2 Report

Authors explained that there is no non-keloid controls. Other comments could not be fully addressed. Could the authors perform IHC staining for the specific markers of mast cells with the extra slides of these keloid tissues? In addition, information for the Figures (legends) and Methods should be more detailed. 
